# The Role and Mechanism of GSDME-Dependent Pyroptosis in Cochlear Marginal Cells Injury by Cisplatin

**DOI:** 10.3390/biomedicines13071680

**Published:** 2025-07-09

**Authors:** Wenyang Lei, Wenting Yu, Ting Li, Wei Tang, Shimin Zong, Hongjun Xiao

**Affiliations:** 1Department of Otorhinolaryngology-Head and Neck Surgery, Union Hospital, Tongji Medical College, Huazhong University of Science and Technology, Wuhan 430022, China; m202175888@alumni.hust.edu.cn (W.L.); ywt_ent@hust.edu.cn (W.Y.); m202376203@hust.edu.cn (T.L.); 18570085383@163.com (W.T.); 2Institute of Otorhinolaryngology-Head and Neck Surgery, Tongji Medical College, Huazhong University of Science and Technology, Wuhan 430022, China; 3Hubei Province Clinical Research Center for Deafness and Vertigo, Wuhan 430022, China

**Keywords:** cisplatin, stria vascularis, marginal cells, pyroptosis, GSDME

## Abstract

**Background:** Elucidating the mechanisms underlying cisplatin-induced ototoxicity is critical for the clinical management of hearing loss. While cisplatin is known to penetrate the inner ear via the blood-labyrinth barrier in the stria vascularis, its precise damaging effects on marginal cells (MCs) and subsequent hearing impairment remain incompletely understood. Pyroptosis, a gasdermin-mediated inflammatory cell death pathway, may play a key role. This study investigated the involvement of gasdermin E (GSDME)-dependent pyroptosis in cisplatin-induced injury to MCs. **Methods:** An in vitro cisplatin-induced pyroptosis model was established in MCs. GSDME expression was downregulated using small interfering RNA (siRNA), and caspase-3 activity was inhibited pharmacologically. The critical threshold for pyroptosis induction was determined to be 5 μmol/L cisplatin exposure for 24 h, which activated the caspase-3/GSDME signaling pathway. **Results:** Cisplatin treatment upregulated GSDME and caspase-3 expression in MCs. Both inhibition of GSDME and pharmacological blockade of caspase-3 significantly attenuated cisplatin-induced cellular damage. Notably, caspase-3 suppression reduced GSDME expression, suggesting a positive regulatory relationship between these mediators. **Conclusions:** GSDME-mediated pyroptosis plays a pivotal role in cisplatin-induced marginal cell injury, with caspase-3 acting as an upstream regulator of GSDME expression. These findings provide a mechanistic foundation for developing novel therapeutic strategies against cisplatin ototoxicity.

## 1. Introduction

Ototoxicity, a common drug-induced adverse reaction, triggers hearing loss and vestibular dysfunctions, significantly compromising patients’ quality of life and exacerbating the burden on public health systems [1,2,3,4]. As a cornerstone in anticancer pharmacology and the most widely used platinum-based chemotherapeutic agent, cisplatin is intensively administered for managing various solid tumors [5,6,7,8]. Nonetheless, its association with ototoxicity is well-documented; the incidence of hearing loss post-cisplatin administration ranges from 40% to 60%, with roughly 20% of patients experiencing severe or profound degrees of hearing impairment [9,10]. The ramifications of this ototoxicity include cognitive deficits, stagnation or regression in speech and language developmental milestones, and social incapacitation, which in turn restrict the drug’s clinical utility [11,12,13,14,15]. Despite its clinical importance, the pathogenesis of cisplatin-induced hearing loss remains elusive, and the repertoire of clinical interventions to counteract or treat cisplatin’s ototoxic effects is currently inadequate. Consequently, there is an exigent demand for research aimed at delineating the precise mechanisms whereby cisplatin gains access to and affects the inner ear.

Cisplatin exerts its ototoxic effects predominantly by impairing the structural integrity and functional capacity of the inner ear, specifically targeting the stria vascularis, spiral ganglion, and the organ of Corti [16,17,18]. While the extant literature has predominantly concentrated on the mechanisms underlying cisplatin-induced damage to the outer hair cells of the organ of Corti [19,20], the stria vascularis has emerged as a significant area of interest due to its distinctive physiological attributes and functions [21,22,23,24]. The principal route of cisplatin entry into the inner ear from the bloodstream is via the blood-labyrinth barrier within the stria vascularis, where it swiftly permeates the endolymphatic fluid, amassing within the cochlea and culminating in inner ear injury [25,26,27]. The ion transport mechanisms, mediated by Na+, K+ ATPase pumps and Na+, K+, Cl− cotransporter channels, are predominantly situated in the marginal cell layer of the stria vascularis [28]. These components are instrumental in maintaining the ionic homeostasis of the endolymph and in the establishment of the cochlear endocochlear potential [29]. From these observations, we postulate that the marginal cells are the principal functional cellular constituents of the stria vascularis [30]. Furthermore, the presence of Copper Transporter 1 (CRT1) and Organic Cation Transporter 2 (OCT2) proteins, pivotal in cisplatin transport, in the marginal cells of the stria vascularis, intimates that these cells may serve as a regulatory gateway for cisplatin entry into the inner ear [31,32,33,34]. In conclusion, the marginal cells are considered to be instrumental in the trafficking of cisplatin into the inner ear. Consequently, an in-depth exploration of the mechanisms underlying cisplatin-induced damage to the marginal cells of the stria vascularis could offer a promising avenue for the development of strategies aimed at attenuating the ototoxic sequelae of cisplatin.

Cisplatin is characterized by its pro-inflammatory properties, which become more pronounced with escalating dosages and extended durations of administration [35,36]. This escalation leads to an increased release of inflammatory cytokines within the cochlea. The dysregulated release of these inflammatory mediators may precipitate cell death, cochlear damage, and ultimately, hearing loss [37,38,39]. Pyroptosis, an emerging form of regulated cell death, is distinguished by the release of a plethora of inflammatory factors consequent to plasma membrane rupture, triggering a robust inflammatory reaction [40,41]. Indeed, this characteristic is unique to pyroptosis, clearly distinguishing it from the process of apoptosis [42]. This process is predominantly orchestrated by the gasdermin family of proteins [43,44,45]. Notably, gasdermin D (GSDMD), identified as the inaugural member of this family with the capacity to form pores in the plasma membrane and execute pyroptosis, is considered the effector of this cell death program [46,47]. Investigations have revealed that cisplatin can inflict damage on the marginal cells of the stria vascularis in rats through the activation of the NLRP3 inflammasome, thereby inducing GSDMD-dependent pyroptosis [48]. GSDME, a protein within the same family as GSDMD, also possesses pore-forming activity at its GSDME-N terminus, enabling it to induce pyroptosis [49,50,51,52]. Moreover, GSDME’s role extends beyond being an independent genetic cause of dominant deafness; it is intimately associated with sensorineural hearing loss [53,54]. This dual function has led to the hypothesis that GSDME may be a pivotal target in cisplatin-induced damage to marginal cells of the stria vascularis, although its precise role in this context remains to be elucidated [54,55,56]. Accordingly, this study is designed to delineate the role and significance of GSDME-dependent pyroptosis in cisplatin-induced damage to marginal cells of the stria vascularis. By establishing an in vitro model of cisplatin-induced pyroptosis in rat marginal cells, this research endeavors to uncover new avenues for clinical prevention and therapeutic intervention for cisplatin-associated hearing loss.

## 2. Materials and Methods

### 2.1. Antibodies and Reagents

Antibodies of GSDME and GSDMD for immunofluorescence staining were purchased from proteintech in Wuhan, China. Antibodies of GSDME for Western blot were purchased from Abcam (Cambridge, MA, USA). Antibodies of caspase-3 were purchased from proteintech in Wuhan, China. Antibodies of β-actin were purchased from Servicebio in Wuhan, China. The secondary antibodies including horseradish peroxidase (HRP)-conjugated goat anti-rabbit were purchased from Servicebio in Wuhan, China. The fluorochrome-conjugated secondary antibody and 4’,6-diamidino-2-phenylindole (DAPI) were purchased from AntGene in Wuhan, China.

Reagents were purchased as followings: Cisplatin (2 mL:10 mg) from Yun Nan Phytopharmaceutical Co., LTD (Kunming, China); 0.1% type II collagenase from Sigma-Aldrich (St. Louis, MO, USA); Epithelial Cell Medium-animal (#4131) from ScienCell (Carlsbad, CA, USA); CCK-8 kit (HY-K0301) and caspase-3 inhitor Z-DEVD-FMK (210344-95-9) from MCE (Monmouth Junction, NJ, USA); FITC Active caspase-3 Apoptosis Kit (550480) and FITC annexin V appotosis kit I (556547) from BD Biosciences (Franklin Lakes, NJ, USA); Opti-MEM (31985070) and DEME (21063029) from gibco (Grand Island, NY, USA); Lipofectamine 3000 (L3000001) from Invitrogen (Carlsbad, CA, USA); LDH Cytotoxicity Assay Kit (C0016) from Beyotime (Shanghai, China); TRIzol was purchased from Nanjing Vazyme Biotech Co., Ltd. (Nanjing, China); Tris-glycine SDS-PAGE SWE High resolution fast electrophoresis buffer (G2081), Rapid transfer buffer (G2028), Electron microscopy fixed liquid (G1102), and Ripa cracking liquid (G2002) from Servicebio (Wuhan, China); RNA isolater Total RNA (R401), reverse transcriptase (R021) and chamQ SYBR qPCR Master Mix (Q311) from Vazyme (Nanjing, China).

### 2.2. Primary MC Culture and Identification

Neonatal Sprague-Dawley (SD) rats aged 48 h postpartum were selected for this study. All experimental animals were housed in the Animal Breeding Center of the Tongji Medical College, Huazhong University of Science and Technology, under sterile conditions. Healthy pups were selected for subsequent experiments at 48 h after birth. The lateral wall of the cochlear stria vascularis was meticulously dissected and isolated. Subsequently, the tissue underwent enzymatic digestion using collagenase for a period ranging from 30 to 60 min. Following digestion, the tissue was centrifuged, and the pellet obtained was resuspended in EpiCM-animal medium. This suspension was then seeded onto cell culture plates and incubated at a temperature of 37 °C within a humidified atmosphere comprising 5% CO_2_. The medium was replaced every 24 h, with cells being gently washed with sterile phosphate-buffered saline (PBS) before each medium renewal. After a cultivation period of 3 to 4 days, the cells were prepared for subsequent experimental manipulations. Immunofluorescence staining confirmed CK-18 expression in cluster-growing primary MCs [48].

### 2.3. Cell Viability Assay

MCs, cultured in 96-well plates, were exposed to a gradient of cisplatin concentrations ranging from 0 to 100 μM. Cell viability at distinct time intervals and across the spectrum of concentrations was evaluated utilizing the CCK-8 detection assay kit, strictly adhering to the protocol provided by the manufacturer. The absorbance of each concentration group was quantified at a wavelength of 490 nm.

### 2.4. Immunofluorescence Staining

MCs were fixed using a 4% paraformaldehyde solution for 40 min at a temperature of 37 °C, followed by three successive rinses with PBS. Permeabilization was achieved using a 5% Triton X-100 solution for a duration of 15 min, after which the cells were blocked with normal donkey serum for 30 min at room temperature. The cells were then incubated with the primary antibody at 4 °C for a period of 18 h. Following a series of washes with PBS containing 0.1% Tween-20 (PBST), the cells were incubated with a fluorescently labeled secondary antibody for a period of 1 h in the dark. Subsequently, the cells were counterstained with DAPI for a period of 10 min to visualize the nuclei. Ultimately, the cells were examined using a laser scanning confocal microscope to assess cellular morphology and antigen localization.

### 2.5. Small Interfering RNA Transfection

The transfection sequences for GSDME and GSDMD were obtained from Tsingke Biotechnology Co., Ltd. (Beijing, China). Peripheral marginal cells were seeded into 6-well plates and incubated for 48 h. Lyophilized siRNA (2.5 nmol) was briefly centrifuged for 30 s and then reconstituted in 125 μL of RNase-free water to prepare a 20 μM stock solution. Subsequently, a transfection solution containing 5 μL small interfering RNA (siRNA) was prepared according to the manufacturer’s instructions, utilizing Lipofectamine 3000 and Opti-MEM transfection reagent. This solution was combined with serum-free Dulbecco’s Modified Eagle Medium (DMEM), incubated for 20 min to allow for complex formation, and then applied to the cells for a further 7 h incubation period. The efficacy of siRNA-mediated knockdown was determined by immunofluorescence staining and quantitative real-time polymerase chain reaction (qPCR) analysis.

### 2.6. Western Blot Analysis

Cells were lysed using Radio-Immunoprecipitation Assay (RIPA) lysis buffer for a duration of 20 min. Following lysis, the lysate was centrifuged at 12,000 revolutions per minute (rpm) for 15 min at 4 °C using an ultracentrifuge to isolate the supernatant. The supernatant was subsequently mixed with a protein loading buffer at a volume ratio of 1:4 and heated in a metal bath at 95 °C for 15 min to denature the proteins. SDS-polyacrylamide gel electrophoresis (SDS-PAGE) was then conducted to resolve the proteins, which were subsequently transferred onto a polyvinylidene fluoride (PVDF) membrane. The membrane was incubated with specific primary antibodies directed against the protein of interest and a loading control antibody for 18 h at 4 °C. After thorough washing with TBST, the membrane was incubated with horseradish peroxidase (HRP)-conjugated secondary antibodies for 1 h. The protein bands were visualized using an enhanced chemiluminescence (ECL) detection kit, and the grayscale value ratio of the protein of interest to the loading control was statistically analyzed to quantify the relative protein expression levels. All bands were analyzed for grayscale values using ImageJ (v1.53, NIH, USA) and statistically processed with GraphPad Prism 9.0 (GraphPad Software).

### 2.7. LDH Release Assay

The release of lactate dehydrogenase (LDH) from marginal cells was evaluated using a standardized assay protocol as recommended by the manufacturer. The supernatant obtained from the cell culture was subjected to LDH activity measurement, with absorbance readings recorded at wavelengths of 450 nm and 630 nm. Upon completion of data acquisition, the absorbance values were processed to ascertain the relative LDH release, which serves as an indicator of cell membrane integrity and cytotoxicity.

### 2.8. Real-Time PCR

Total RNA was isolated from marginal cells using TRIzol reagent. Subsequently, the extracted RNA underwent reverse transcription to synthesize complementary DNA (cDNA) using a commercially available kit. For reverse transcription, a reaction mixture was prepared by combining 4 μL of RNA, 4 μL of DEPC-treated water, and 2 μL of 5× reaction buffer (final volume: 10 μL). Quantitative real-time polymerase chain reaction (qRT-PCR) was performed on the qRT-PCR system to assess the relative mRNA expression levels. The relative quantification was achieved using the comparative cycle threshold (ΔΔCT) method. The primer sequences employed in this study are detailed in Appendix A.

In this study, six candidate reference genes were initially selected and evaluated. Using the NormFinder v0.953, we analyzed their expression stability in the lateral wall of the stria vascularis in SD rats. The analysis indicated that GAPDH had the most stable expression profile among the candidates. Therefore, GAPDH was selected as the reference gene for subsequent qRT-PCR experiments. The sequences of the six reference genes and the detailed analysis results are presented in Appendix A and Appendix A. The raw data included in the Appendix A.

### 2.9. Flow Cytometry

Caspase-3 activity and the proportion of cell death were evaluated using a caspase-3 flow cytometry detection kit and propidium iodide (PI), strictly adhering to the manufacturer’s guidelines. The procedure commenced with the treatment of marginal cells, followed by enzymatic digestion using trypsin devoid of EDTA for a duration of 3 min to detach the cells from the culture surface. The cells were then subjected to centrifugation to pellet the cellular material. Afterward, the cell pellet was resuspended and washed with sterile PBS to remove any residual trypsin. The cells were subsequently fixed and permeabilized using BD Cytofix/Cytoperm™ (Franklin Lakes, NJ, USA) solution for 20 min at room temperature, facilitating the subsequent antibody binding. The cells were then incubated with a specific caspase-3 antibody in a light-protected environment for 30 min to allow for antibody–antigen interaction. Following the incubation, the cells were resuspended in a mixture containing BDwash and PI to stain the cells. Ultimately, the stained marginal cells were subjected to flow cytometric analysis to quantify the population of cells that were double-positive for caspase-3 and PI, thereby providing a measure of both caspase-3 activity and cell death. All flow cytometry data were analyzed using FlowJo v10.0 (BD Biosciences, USA) and statistically processed with GraphPad Prism 9.0 (GraphPad Software, San Diego, CA, USA).

### 2.10. Cell Ultrastructure Observation Under Electron Microscope

Marginal cells, after undergoing experimental manipulation, were fixed with an electron microscopy (EM) fixation solution for a duration of 30 min to preserve their structural integrity. The processed cell samples were subsequently subjected to both scanning electron microscopy (SEM) and transmission electron microscopy (TEM) to examine their morphological features and internal structures.

Sample Processing for SEM Analysis: The marginal cells were processed according to experimental groups, with cisplatin-treated cells washed twice in PBS before fixation in EM fixative for 5 min at room temperature (protected from light). Cells were gently scraped in one direction, pelleted by centrifugation (1000 rpm, 4 °C, 3 min), and resuspended in fresh fixative for 30 min before 4 °C storage. Following dehydration through an ethanol series and acetone, samples were infiltrated with acetone/812 embedding medium (1:1, 37 °C, 3 h), then transitioned to a 2:1 medium/acetone mixture overnight before pure medium embedding and polymerization. After curing at 60 °C for 48 h, 70 nm sections were collected on Formvar-coated grids, sequentially stained with uranyl acetate and lead citrate (10 min each, with ethanol/DEPC water rinses), and examined by scanning electron microscopy to document ultrastructural changes.

### 2.11. Statistical Analysis

All quantitative data are reported as the mean ± standard deviation (SD), calculated from a minimum of three independent experiments to ensure the robustness of the findings. The statistical analyses were performed using GraphPad Prism software. A *p*-value of less than 0.1 was established as the threshold for statistical significance, indicating that the observed differences were unlikely to occur by chance.

## 3. Results

### 3.1. Cisplatin Induced Pyroptosis in MCs

Prior research has defined pyroptosis through the hallmarks of compromised plasma membrane integrity, intracellular vacuolization, and the concurrent release of LDH. In our study, MCs were exposed to a range of cisplatin concentrations (0, 5, 10, 20, 50, 100 μmol/L) over periods of 24, 36, and 48 h. Cell viability was evaluated using the CCK-8 assay, and the findings (Figure 1A) illustrated a progressive decrease in viability correlated with both cisplatin concentration and exposure duration. The calculated half-maximal inhibitory concentrations (IC50) for the three time points were 9.185 μmol/L, 4.224 μmol/L, and 3.587 μmol/L, respectively. However, upon examination with an inverted microscope following 36 h of cisplatin treatment, the MCs demonstrated suboptimal growth and altered morphology. Therefore, a treatment duration of 24 h was chosen for subsequent experiments to maintain cellular integrity.

Following a 24 h treatment with 0, 5, and 10 μM cisplatin concentrations, LDH release into the supernatant was quantified (Figure 1B). The LDH release percentages for the respective cisplatin groups were 7.87%, 20.06%, and 21.38%, respectively, suggesting a concentration-dependent pattern of LDH release. Subsequent analysis of cells treated with 5 μM cisplatin for 24 h using SEM and TEM revealed ultrastructural alterations. SEM observations (Figure 1C,D) highlighted membrane damage, cellular edema with cytoplasmic leakage, and pronounced disruption of microvilli in cisplatin-exposed cells. TEM analysis (Figure 1E,F) further confirmed the loss of membrane integrity and the formation of characteristic large vacuoles, distinguishing the treated cells from the control group, which exhibited an intact membrane and preserved cell health. Collectively, these observations indicate that cisplatin can trigger pyroptosis in MCs.

### 3.2. Cisplatin Upregulates the Expression of Pyroptosis Related Molecules in MCs

In an effort to unravel the molecular mechanisms of cisplatin-induced pyroptosis in marginal cells, the cells were subjected to cisplatin at concentrations of 5 μM and 10 μM for a duration of 24 h. The expression profiles of GSDME and caspase-3 at the mRNA and protein levels were evaluated using qRT-PCR and immunofluorescence staining, respectively. As illustrated in Figure 2, the expression of GSDME at the transcriptional and translational levels was most significantly upregulated in response to the 5 μM cisplatin treatment. Similarly, caspase-3 mRNA expression was most elevated at this concentration, with a corresponding increase in protein expression that correlated with the dose of cisplatin administered. These observations suggest that the 24 h treatment with 5 μM cisplatin could be a pivotal time point for initiating pyroptosis in MCs. Moreover, the results indicate that cisplatin may induce pyroptosis in these cells by modulating the caspase-3/GSDME signaling pathway.

### 3.3. Down-Regulation of GSDME Alleviated the Cisplatin-Induced Pyroptosis in MCs

In this investigation, we employed siRNA to suppress GSDME expression in marginal cells. The proportion of caspase-3-positive cells exhibiting plasma membrane compromise (Q4 quadrant) was evaluated using flow cytometry, as illustrated in Figure 3A,B. The siRNA-GSDME + cisplatin, siRNA-NC + cisplatin, cisplatin-only, and control groups displayed double-positive cell percentages of 11.33%, 19.13%, 20.13%, and 1.39%, respectively. These results indicate a notable increase in double-positive cells upon cisplatin treatment compared to the control, with a decrease observed following GSDME suppression relative to the cisplatin + siRNA-NC group.

Measurement of LDH release into the supernatant of cell cultures (Figure 3E) revealed LDH release percentages of 6.87%, 17.86%, 19.11%, and 6.55% for the siRNA-GSDME + cisplatin, siRNA-NC + cisplatin, cisplatin-only, and control groups, respectively. This data suggests a significant decrease in LDH release in the siRNA-GSDME + cisplatin group compared to the siRNA-NC + cisplatin group, alongside a significant increase in LDH release in cisplatin-treated groups relative to the control.

Western blot analysis (Figure 3C) confirmed elevated levels of full-length GSDME (GSDME-FL), cleaved GSDME (GSDME-N), caspase-3, and cleaved caspase-3 proteins in cisplatin-treated groups compared to the control. Within the siRNA-GSDME + cisplatin group, the levels of GSDME-FL and GSDME-N were diminished compared to the siRNA-NC + cisplatin group, with no significant variation in caspase-3 and cleaved caspase-3 protein levels. Immunofluorescence staining (Figure 3D,F,H) corroborated the increased expression of GSDME and caspase-3 in response to cisplatin treatment compared to the control. Conversely, in the siRNA-GSDME + cisplatin group, GSDME expression was downregulated compared to the siRNA-NC + cisplatin group, while caspase-3 expression remained relatively stable.

In conclusion, these findings suggest that GSDME plays a role in mediating cisplatin-induced pyroptosis in MCs, yet it does not modulate the expression levels of caspase-3.

### 3.4. Down-Regulation of GSDME Alleviated the Pyroptotic Morphological Changes in MCs After Cisplatin Treatment

Cell swelling, plasma membrane rupture, and the formation of large vacuoles are hallmark morphological changes indicative of pyroptosis. As shown in Figure 4A, TEM images disclosed that MCs from the cisplatin-treated group and the siRNA-NC + cisplatin group presented with cell swelling, plasma membrane disruption accompanied by cytoplasmic leakage, and the formation of pyroptotic body. In contrast, cells in the control group and the siRNA-GSDME + cisplatin group retained their structural integrity, with distinct surface microvilli visible. SEM provided further insights (Figure 4B), illustrating that the plasma membranes of marginal cells in the cisplatin-treated and siRNA-NC + cisplatin groups were characterized by spiculated appearances, multiple rupture sites, and the presence of large vacuoles, which are indicative of cytoplasmic dissolution and vacuolation. Conversely, cells in the control group and the siRNA-GSDME + cisplatin group exhibited plasma membranes that were intact and smooth. These findings imply that the downregulation of GSDME may mitigate the pyroptosis-like morphological alterations induced by cisplatin in MCs.

### 3.5. Inhibition of Caspase-3 Attenuates Cisplatin-Induced Pyroptosis in MCs

The current study has established that cisplatin treatment leads to upregulation of caspase-3 and GSDME in MCs, yet the relationship between these two factors is not fully understood. To investigate their interplay, MCs were exposed to the caspase-3 inhibitor Z-DEVD-FMK, and the presence of caspase-3-positive cells with compromised plasma membranes was evaluated using flow cytometry, as illustrated in Figure 5A. The Z-DEVD + cisplatin, Z-DEVD-only, cisplatin-only, and control groups exhibited double-positive cell percentages of 2.5%, 1.11%, 3.98%, and 0.9%, respectively. These findings indicate a decrease in the number of double-positive cells in the Z-DEVD + cisplatin group relative to the cisplatin-only group, with an increase in comparison to the Z-DEVD-only and control groups.

Measurement of LDH release into the supernatant of cell cultures (Figure 5B) revealed LDH release percentages of 11.00%, 8.38%, 16.09%, and 7.94% for the Z-DEVD + cisplatin, Z-DEVD-only, cisplatin-only, and control groups, respectively. This suggests a reduction in LDH release in the Z-DEVD + cisplatin group compared to the cisplatin-only group, and an increase relative to the Z-DEVD-only group.

Western blot analysis (Figure 5C) demonstrated reduced protein expression of full-length GSDME (GSDME-FL), cleaved GSDME (GSDME-N), and cleaved caspase-3 in the Z-DEVD + cisplatin group when compared to the cisplatin-only group. TEM observations (Figure 5H) indicated that the Z-DEVD + cisplatin group had less severe plasma membrane damage than the cisplatin-only group but more pronounced damage than the control group. The number of pyroptotic bodies was reduced compared to the cisplatin-only group but increased compared to the control group. The quantity of pyroptotic bodies showed a significant decrease relative to the cisplatin-only group, while remaining elevated compared to the control group. In contrast, the control and Z-DEVD-only groups displayed intact plasma membranes and maintained healthy cellular morphology. SEM (Figure 5I) showed that the Z-DEVD + cisplatin group had a lesser extent of cellular damage than the cisplatin-only group but more significant damage than the Z-DEVD-only group.

In conclusion, these results suggest that inhibition of caspase-3 activity can ameliorate cisplatin-induced pyroptosis in marginal cells.

### 3.6. Caspase-3 Induces GSDME-Mediated Pyroptosis in MCs

To delve deeper into the downstream effectors of caspase-3 in response to cisplatin treatment, we employed small interfering RNA (siRNA) to suppress the expression of GSDMD in marginal cells. Subsequently, we assessed the expression levels of GSDMD and GSDME following the inhibition of caspase-3. Following GSDMD knockdown, LDH release into the cell culture supernatant was quantified (Figure 6A). The si-GSDMD + cisplatin, si-NC + cisplatin, cisplatin-only, and control groups exhibited LDH release rates of 7.62%, 18.90%, 24.59%, and 8.65%, respectively. The observed reduction in LDH release for the siRNA-GSDMD + cisplatin group relative to the siRNA-NC + cisplatin group implies that the downregulation of GSDMD may mitigate cisplatin-induced plasma membrane disruption in marginal cells. Real-time PCR and immunofluorescence data, presented in the subsequent figures, reveal that the expression of GSDME in the Z-DEVD + cisplatin group was diminished compared to the cisplatin-only group, whereas the expression of GSDMD remained largely unchanged compared to the cisplatin-only group. Collectively, these results suggest that caspase-3, under cisplatin treatment, activates GSDME rather than GSDMD in MCs.

## 4. Discussion

Cisplatin selectively accumulates in the stria vascularis, causing progressive structural and functional damage. Such impairments encompass alterations in the permeability and integrity of the blood-labyrinth barrier, the induction of ultrastructural modifications in MCs, and a significant decrease in cochlear endocochlear potential [57,58,59,60,61]. Critically, key ion channels like KCNQ1/KCNE1 and transporters such as Na+/K+-ATPase are highly enriched in MCs [62,63,64,65]. Thus, cisplatin’s targeted damage to MCs may be central to ototoxicity. The findings of this study underscore that cisplatin triggers pyroptosis in MCs via the activation of the caspase-3/GSDME signaling cascade (Figure 7). This pathway could represent a potential mechanism underlying strial marginal cell injury induced by cisplatin. These insights may pave the way for novel clinical research directions focused on attenuating the ototoxic sequelae of cisplatin treatment. This pathway may serve as a potential mechanism underlying cisplatin-induced injury to marginal cells of the stria vascularis. These findings could offer a novel direction for clinical research focused on alleviating cisplatin ototoxicity—specifically by targeting and inhibiting the caspase-3/GSDME signaling pathway, in conjunction with cochlear vascular labyrinth barrier protectants to achieve synergistic otoprotective effects.

The elucidation of the GSDM protein family has significantly advanced our understanding of pyroptosis mechanisms [66,67,68,69]. GSDME-dependent pyroptosis represents a distinct pro-inflammatory cell death pathway characterized by plasma membrane rupture and subsequent release of inflammatory mediators and lactate dehydrogenase (LDH) [70,71,72,73,74,75,76]. In our study, cisplatin-induced injury to MCs was observed to be contingent upon both temporal and dosage factors, concomitant with elevated expression levels of GSDME and its GSDME-N terminal fragment. GSDME knockdown attenuated cisplatin-induced plasma membrane disruption in MCs, reduced LDH release into the culture supernatant, and decreased GSDME-N fragment generation. Electron microscopic examination disclosed that the downregulation of GSDME was correlated with attenuated plasma membrane damage in MCs and a reduced frequency of pyroptotic bodies. These observations collectively suggest that the activation of GSDME and the concomitant formation of the GSDME-N fragment could be integral to the mechanism through which cisplatin triggers pyroptosis in MCs.

Notably, increasing cisplatin concentrations elicit a biphasic GSDME expression profile in MCs, featuring an initial upregulation followed by downregulation, which parallels progressive LDH release in culture supernatants. This observation suggests the potential involvement of additional intracellular pathways that may precipitate pyroptosis, culminating in cellular rupture and sustained LDH release. Previous studies have implicated GSDMD in cisplatin-induced MC pyroptosis, possibly via NLRP3 inflammasome-mediated caspase-1 activation [48]. The interplay between GSDME and GSDMD during the induction of marginal cell pyroptosis, the possibility of shared regulatory targets such as the NLRP3 inflammasome, and the existence of other signaling molecules that could induce pyroptosis, all warrant rigorous further investigation. Moreover, contemporary research has unveiled that chemotherapeutic agents can induce palmitoylation at the C-terminal of GSDME, precipitating the dissociation of the GSDME-N fragment [77]. The ensuing interaction between the N-terminal and C-terminal fragments has been linked to an incremented frequency of pyroptotic cells [78,79,80]. Consequently, it merits exploration whether cisplatin might induce pyroptosis in MCs through the palmitoylation of GSDME’s C-terminal, the reciprocal influence of the palmitoylated GSDME-C fragment and the GSDME-N fragment, and the potential role of other palmitoylation sites in modulating cisplatin-induced pyroptosis in MCs as avenues for future research.

Caspase-3, traditionally recognized as a pivotal effector in the execution of apoptosis, has emerged as a significant player in the orchestration of pyroptosis through the regulation of GSDME, as evidenced by a growing body of research [81,82,83,84,85]. In our investigation, the suppression of caspase-3 using Z-DEVD-FMK was observed to diminish the incidence of cisplatin-induced marginal cell rupture, coincident with a downregulation in GSDME expression and a curtailed release of LDH in the supernatant of the marginal cell culture medium. This has led to the hypothesis that the caspase-3/GSDME signaling axis may function as a regulatory ‘switch,’ modulating the transition between apoptotic and pyroptotic pathways within MCs under cisplatin influence, with the latter being triggered by the activation of this pathway. Empirical evidence has corroborated the role of caspase-3 in the mitochondrial-mediated demise of hair cells, with the application of Z-DEVD-FMK significantly attenuating cisplatin’s deleterious effects. This implicates that caspase-3-mediated pyroptosis may be integral to cisplatin’s detrimental impact on MCs, with GSDME potentially serving as a nodal point of regulation between these death modalities, hinting at a dynamic interplay and reciprocal modulation. Furthermore, it has been noted that Z-DEVD-FMK, in its capacity to inhibit caspase-3, concurrently suppresses the expression of caspase-7 and caspase-8 [86,87]. The involvement of these caspases in cisplatin-induced marginal cell damage, the existence of alternative signaling pathways that may inhibit caspase-3 expression, and the intricate regulatory dynamics among these signaling components warrant thorough elucidation. Unraveling the precise mechanisms of cell death engendered by cisplatin in marginal cells and identifying the proteins implicated therein could unveil novel therapeutic targets and chart new research trajectories for the amelioration of cisplatin’s ototoxic consequences.

The anatomical distinctiveness of the MCs within the stria vascularis presents a significant challenge for targeted genetic interventions, thereby complicating the execution of animal studies. To address this, the current experiment utilized primary rat marginal cells from the stria vascularis for in vitro cultivation. While primary cells offer a more authentic representation of cellular physiology and status as compared to cell lines, they are not a panacea for the complexities inherent in animal models. The efficacy of GSDME-dependent pyroptosis in animal models and its consistency with in vitro observations necessitate further empirical validation. The scarcity of tools designed for the precise modulation of GSDME in vivo represents a formidable challenge for forthcoming research endeavors. Moreover, the absence of methodologies for the quantitative assessment of pyroptotic MCs precludes the definitive exclusion of alternative cell death pathways potentially triggered by cisplatin. The precise temporal and concentration-dependent dynamics of such occurrences, the interplay and mutual influence among diverse cell death modalities, and the feasibility of uncovering shared signaling pathways or specific targets for tailored intervention strategies all demand rigorous and extensive investigation.

## Figures and Tables

**Figure 1 biomedicines-13-01680-f001:**
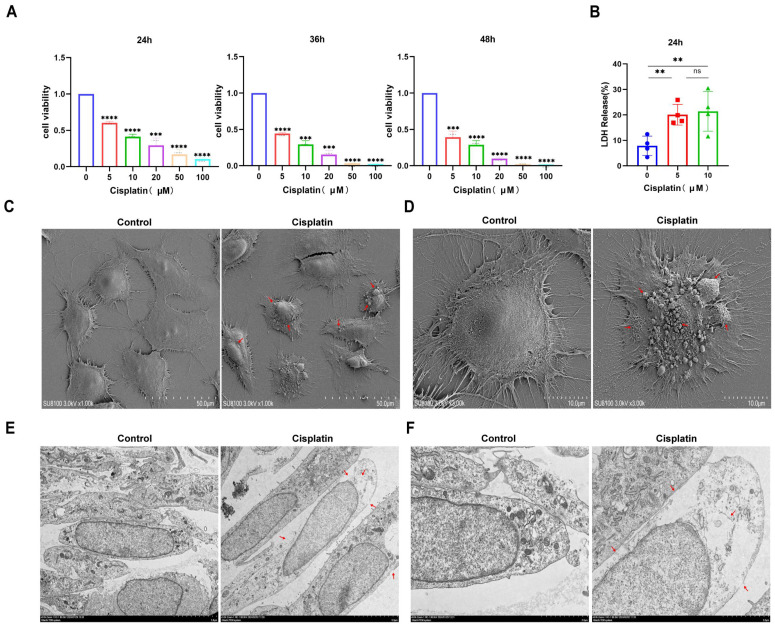
Cisplatin induces pyroptosis in MCs of the vascular wall. (**A**) The cytotoxic effect of cisplatin on MCs is both time- and concentration-dependent, with a decrease in cell viability observed as treatment concentration and duration increase (CCK-8 assay, n ≥ 3 biological replicates per group); (**B**) LDH release assay demonstrating cisplatin-induced membrane rupture (n ≥ 3 biological replicates per group); (**C**) low-magnification SEM images reveal ultrastructural changes in MCs following cisplatin treatment. Scale bar: 50.0 μm (n = 3 biological replicates per group; (**D**) high-magnification SEM images provide a detailed view of the ultrastructural changes in MCs following cisplatin treatment. Scale bar: 10.0 μm (n = 3 biological replicates per group); (**E**) low-magnification TEM images show ultrastructural alterations in MCs after cisplatin exposure. Scale bar: 5.0 μm (n = 3 biological replicates per group; (**F**) high-magnification TEM images offer a closer look at the ultrastructural changes in MCs after cisplatin exposure. Scale bar: 2.0 μm (n = 3 biological replicates per group. Red arrows indicate cisplatin-induced membrane disruption, pyroptotic bodies (characteristic balloon-like protrusions), and characteristic pyroptotic morphology. All experiments were repeated independently three times with consistent results. Statistical significance was determined by one-way ANOVA followed by Tukey’s HSD post hoc test and is indicated as follows: ** *p* < 0.01, *** *p* < 0.001, **** *p* < 0.0001, ns: No significant difference.

**Figure 2 biomedicines-13-01680-f002:**
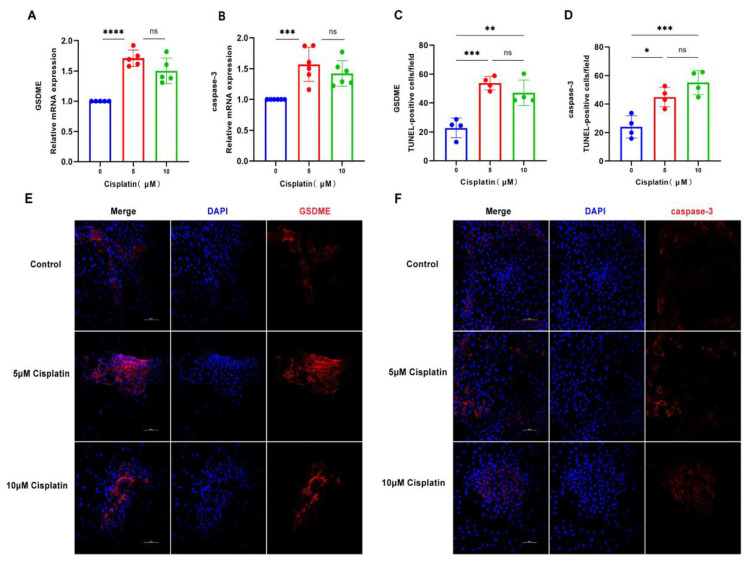
Cisplatin triggers the upregulation of pyroptosis-associated proteins in MCs. (**A**) qRT-PCR analysis of GSDME mRNA levels in MCs treated with cisplatin (5 μM and 10 μM) for 24 h, normalized to GAPDH); (**B**) caspase-3 mRNA levels are also significantly elevated under the same treatment conditions; (**C**–**F**) immunofluorescence staining of GSDME (**C**,**E**) and caspase-3 (**D**,**F**) in MCs after 24 h cisplatin exposure (5 μM and 10 μM). Nuclei were counterstained with DAPI (blue), and target proteins were labeled with Alexa Fluor 647 (red). Scale bars: 100 μm. Quantification of GSDME and caspase-3 fluorescence intensity normalized to DAPI signal in the same region of interest (ROI). Background fluorescence was subtracted using cell-free areas. All data represent n ≥ 3 biological replicates (independent animals), with experiments repeated three times independently. Statistical significance was determined by one-way ANOVA followed by Tukey’s HSD post hoc test and is indicated as follows: * *p* < 0.05, ** *p* < 0.01, *** *p* < 0.001, **** *p* < 0.0001, ns: No significant difference.

**Figure 3 biomedicines-13-01680-f003:**
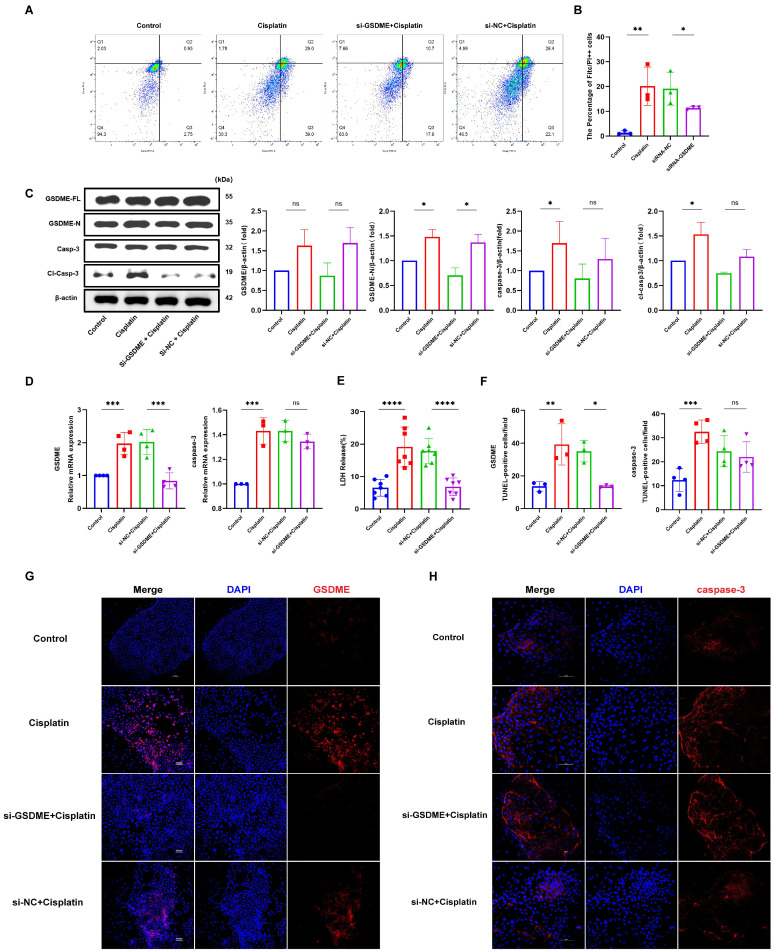
GSDME knockdown attenuates cisplatin-induced pyroptosis in MCs. (**A**,**B**) Cisplatin treatment and transfection with non-targeting siRNA led to a significant increase in the number of caspase-3-positive cells exhibiting plasma membrane disruption compared to the untreated control and si-GSDME + cisplatin groups, respectively; (**C**) Cisplatin treatment resulted in elevated protein expression levels of both GSDME-N and cl-caspase-3, whereas in the si-GSDME + cisplatin group, GSDME-N and cl-caspase-3 protein expression was not significantly different from the si-NC + cisplatin group. (**D**) Comparative analysis of GSDME and caspase-3 mRNA expression levels in edge cells revealed that GSDME knockdown (si-GSDME + cisplatin group) significantly reduced GSDME mRNA expression compared to the si-NC + cisplatin group, while caspase-3 mRNA expression showed no significant differences between these two groups. (**F**,**G**) Comparative immunofluorescence analysis of GSDME and caspase-3 expression in MCs among the si-GSDME + cisplatin, si-NC + cisplatin, cisplatin, and control groups. Nuclei: DAPI (blue); proteins: Alexa Fluor 647 (red). Scale bars: 100 μm (**G**,**H**). Quantification of GSDME and caspase-3 fluorescence intensity normalized to DAPI signal in the same region of interest (ROI). Background fluorescence was subtracted using cell-free areas; (**E**) LDH release into the supernatant of MCs was diminished in the si-GSDME + cisplatin group relative to the si-NC + cisplatin group. All data represent n ≥ 3 biological replicates (independent animals), with experiments repeated three times independently, with statistical significance was determined by one-way ANOVA followed by Tukey’s HSD post hoc test and is indicated as follows: * *p* < 0.05, ** *p* < 0.01, *** *p* < 0.001, **** *p* < 0.0001, ns: No significant difference.

**Figure 4 biomedicines-13-01680-f004:**
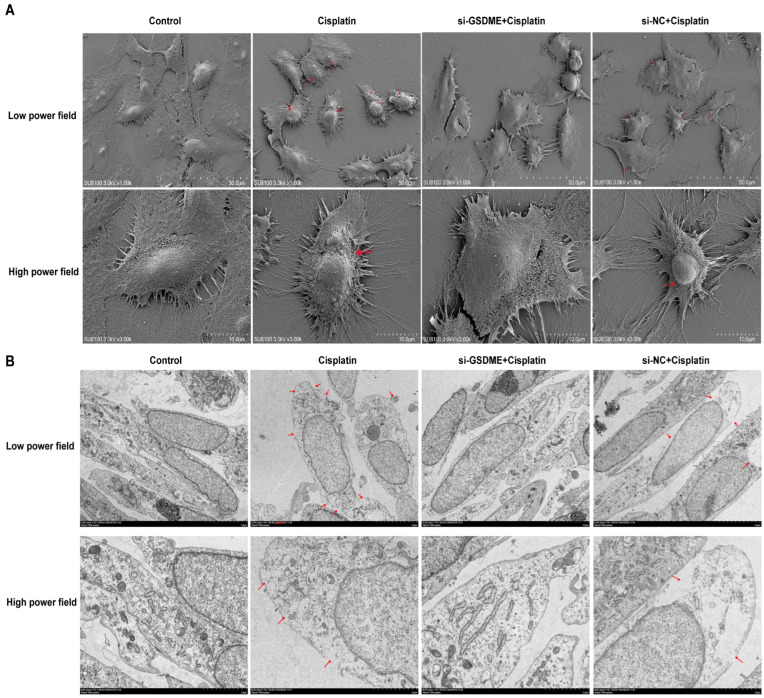
Down-regulation of GSDME alleviates cisplatin-induced morphological changes in MCs. (**A**) SEM analysis of MCs at low (top; scale bar: 50.0 μm) and high (bottom; scale bar: 10.0 μm) magnifications. Red arrows indicate membrane rupture, cytoplasmic leakage and pyroptotic body formation. (**B**) TEM analysis of MCs at low (top; scale bar: 5.0 μm) and high (bottom; scale bar: 2.0 μm) magnifications. Red arrows show plasma membrane spiculation, large vacuoles and cytoplasmic dissolution. All data represent n = 3 biological replicates (independent animals), with experiments repeated three times independently.

**Figure 5 biomedicines-13-01680-f005:**
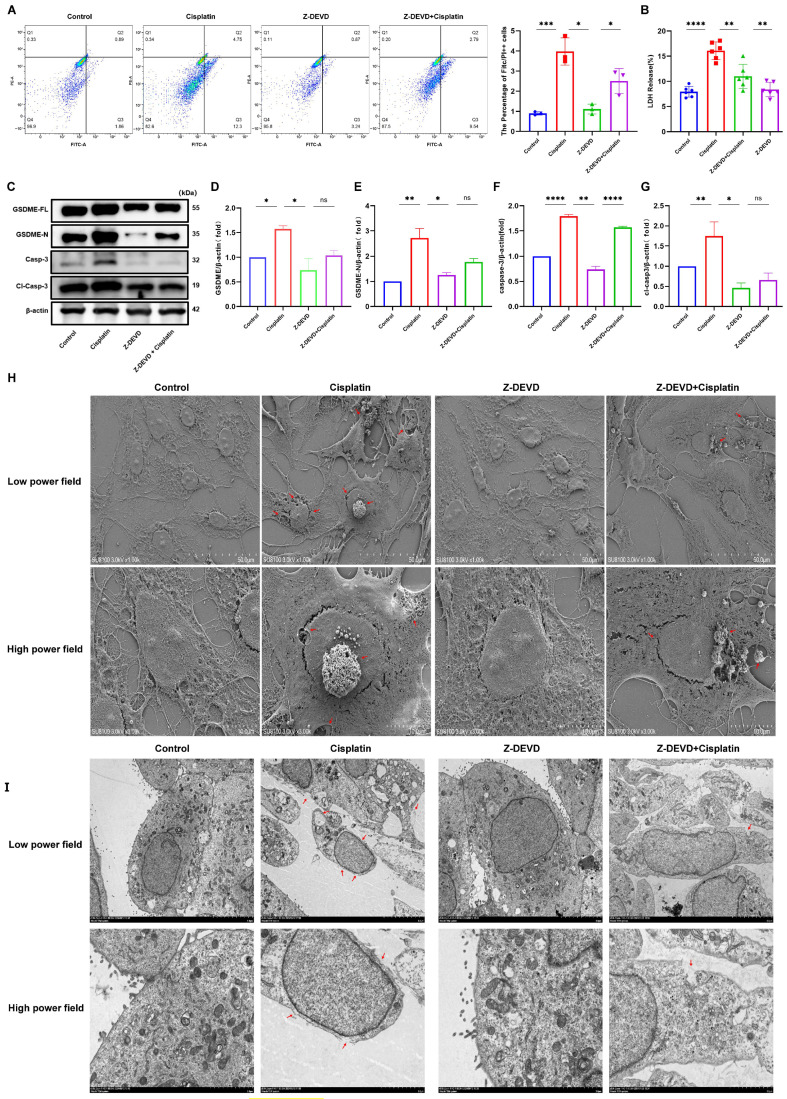
Caspase-3 inhibition mitigates cisplatin-induced pyroptosis in MCs. (**A**) Quantification of caspase-3-positive MCs with membrane disruption shows reduced pyroptosis in Z-DEVD + cisplatin group versus cisplatin group (*p* < 0.001), but increased versus Z-DEVD group (*p* < 0.05); (**B**) LDH release into the supernatant of MCs was decreased in the Z-DEVD + cisplatin group relative to the cisplatin group (*p* < 0.01); (**H**) SEM analysis of MCs at low (top; scale bar: 50.0 μm) and high (bottom; scale bar: 10.0 μm) magnifications. Red arrows indicate membrane rupture, cytoplasmic leakage and pyroptotic body formation. (**I**) TEM analysis of MCs at low (top; scale bar: 5.0 μm) and high (bottom; scale bar: 2.0 μm) magnifications. Red arrows show plasma membrane spiculation, large vacuoles and cytoplasmic dissolution. All data represent n = 3 biological replicates (independent animals), with experiments repeated three times independently. (**C–G**) The cisplatin group showed increased protein expression levels of GSDME-N, caspase-3 and cl-caspase-3 compared to the control group, whereas the Z-DEVD + cisplatin displayed no significant difference in GSDME and caspase-3 protein expression compared to the Z-DEVD group. All data represent n = 3 biological replicates (independent animals), with experiments repeated three times independently. Statistical significance was determined by one-way ANOVA followed by Tukey’s HSD post hoc test and is indicated as follows: * *p* < 0.1, ** *p* < 0.05, *** *p* < 0.01, **** *p* < 0.001, ns: No significant difference.

**Figure 6 biomedicines-13-01680-f006:**
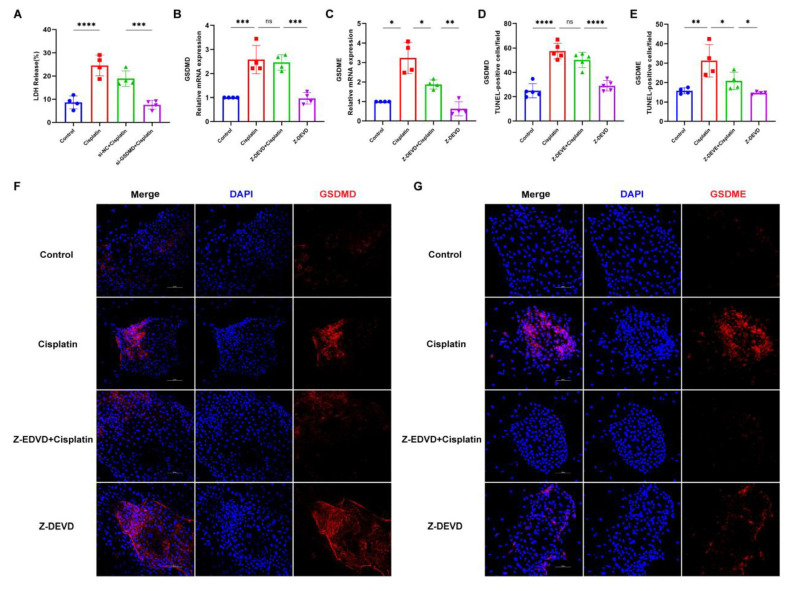
Caspase-3 activates GSDME but not GSDMD to induce MCs pyroptosis following cisplatin treatment. (**A**) LDH release in the supernatant of MCs is compared across the siRNA-GSDMD + cisplatin, siRNA-NC + cisplatin, cisplatin, and control groups; (**B**,**C**) mRNA expression levels of GSDME and GSDMD in marginal cells are analyzed and compared among the Z-DEVD + cisplatin, Z-DEVD, cisplatin, and control groups; (**D**–**G**) immunofluorescence analysis of GSDME and GSDMD expression in MCs across treatment groups (Z-DEVD + cisplatin, Z-DEVD, cisplatin, and control) revealed that caspase-3 inhibitor pretreatment significantly downregulated cisplatin-induced GSDME expression, whereas GSDMD expression remained unchanged between Z-DEVD + cisplatin group and Ciplatin group (nuclei stained blue with DAPI; target proteins labeled red with Alexa Fluor 674). Scale bar: 100 μm. Quantification of GSDMD and GSDME fluorescence intensity normalized to DAPI signal in the same region of interest (ROI). Background fluorescence was subtracted using cell-free areas. All data represent n ≥ 3 biological replicates (independent animals), with experiments repeated three times independently. Statistical significance was determined by one-way ANOVA followed by Tukey’s HSD post hoc test and is indicated as follows: * *p* < 0.05, ** *p* < 0.01, *** *p* < 0.001, **** *p* < 0.0001, ns: No significant difference.

**Figure 7 biomedicines-13-01680-f007:**
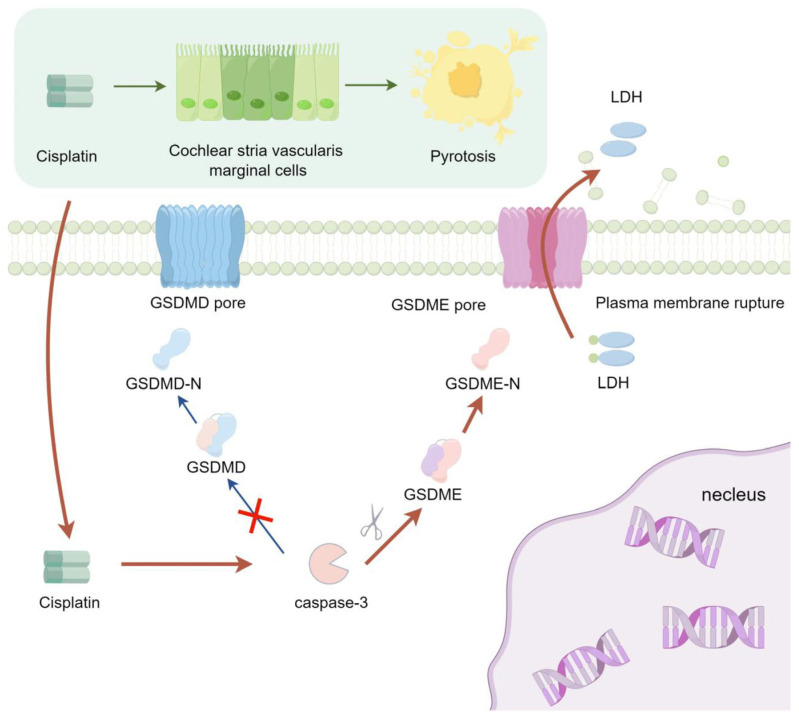
The signaling pathway mechanism by which cisplatin induces pyroptosis in MCs of the stria vascularis in SD rat cochlear.

## Data Availability

The raw data supporting the conclusions of this article will be made available by the authors, without undue reservation.

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
