# Peer review of "The Role and Mechanism of GSDME-Dependent Pyroptosis in Cochlear Marginal Cells Injury by Cisplatin"

_biomedicines, 2025, doi:10.3390/biomedicines13071680_

Round 1
Reviewer 1 Report (Previous Reviewer 1)
Comments and Suggestions for Authors
All the comments have been addressed. The manuscript is recommended for publication.
Author Response
June 13, 2025
Dear Reviewer,
Thank you for your positive feedback on our study. We sincerely appreciate your time and thoughtful review. Wishing you all the best.
Best regards,
Hongjun Xiao
Corresponding Author
Email: xhjent_whxh@hust.edu.cn
Phone: (+86) 13995637098
Reviewer 2 Report (Previous Reviewer 2)
Comments and Suggestions for Authors
The paper has been partially revised, but still contains a number of shortcomings that require additional corrections and refinements.
In the response to the reviews, important technical parameters were not fully considered, including the exact concentrations of proteins and RNA, the number of cells used in the experiments and the sequences of the primers used. These data are necessary for full reproducibility and understanding of the experiments performed.
Also, the abbreviations used are not explained in the paper when they first appear, which makes understanding the content even more difficult, especially for readers who are not directly specialised in the field mentioned.
In my opinion, it is not enough to simply give the same number of animals as control parameters. Quantitative data is needed – a certain number or at least a range of live cells that were actually seeded into each experimental dish. Furthermore, the number of animals used is not explicitly stated anywhere in the text, which further limits the transparency and reproducibility of the experimental design.
The captions are still insufficiently explained; important information necessary for the interpretation of the results shown is missing. It must be ensured that each figure and its legend are understandable without the need for additional context from the rest of the paper.
The features that the authors highlighted in the responses as elements that add to the scientific value of the paper are not included in the text of the manuscript itself. Even when the reference used to characterise the cells is cited in the responses, this information is not included in the text of the paper, further reducing the clarity and completeness of the methodological approach.
The results could be further revised and structured to make the rationale of the research and the main messages that the authors wish to emphasise easier to understand.
Finally, no significant refinement or expansion of the discussion itself was noted. It would be expected that the authors would critically interpret the results obtained in this section, place them in the context of existing knowledge and clearly emphasise the contribution and limitations of their work.
Author Response
June 13, 2025
Dear Reviewer,
Thank you for your letter! We greatly appreciate the opportunity to revise and resubmit our manuscript. We have addressed all reviewers' comments point-by-point. New revisions are highlighted in yellow in the manuscript, while previous revisions remain in green for easy comparison. Supplemental files contain additional materials. Below are our responses to each reviewer comments.
Reviewer:
Comment 1:
[In the response to the reviews, important technical parameters were not fully considered, including the exact concentrations of proteins and RNA, the number of cells used in the experiments and the sequences of the primers used. These data are necessary for full reproducibility and understanding of the experiments performed.]
Response 1:
Thank you for pointing out the shortcomings in our manuscript! We have included the original data for RNA concentration measurements, protein standard curves, and protein quantification in Supplementary Tables 8 and 7 and Supplementary Figure 1 (highlighted in yellow) to ensure scientific reproducibility. The primer sequences used in the experiments are provided in Supplementary Tables 1 and 2.
In this study, we used primary cultures of marginal cells from the lateral wall of the cochlear stria vascularis in SD rat neonates. These cells grow irregularly in patches, making it difficult to standardize cell counts across experiments. However, to ensure scientific rigor, standardization, and reproducibility, we strictly controlled the number of animals per experiment to maintain consistent initial cell quantities. For each experiment, cochlear stria vascularis tissues from 4 neonates were cultured in one 6-well plate. After treatment with different conditions, we performed RNA/protein extraction or immunofluorescence staining to maximize experimental consistency.
Comment 2:
[Also, the abbreviations used are not explained in the paper when they first appear, which makes understanding the content even more difficult, especially for readers who are not directly specialised in the field mentioned.]
Response 2:
We sincerely appreciate your valuable suggestions! As advised, we have standardized all abbreviations throughout the manuscript by using full terms with abbreviations in parentheses at first mention (e.g., "copper transporter 1 (CTR1)") and employing abbreviations consistently thereafter, with all abbreviations and their full terms compiled in Supplementary Table 6 (highlighted in yellow) for reference.
Comment 3:
[In my opinion, it is not enough to simply give the same number of animals as control parameters. Quantitative data is needed – a certain number or at least a range of live cells that were actually seeded into each experimental dish. Furthermore, the number of animals used is not explicitly stated anywhere in the text, which further limits the transparency and reproducibility of the experimental design.]
Response 3:
We sincerely appreciate your valuable comments. In this study, we used primary cultures of irregularly patch-growing marginal cells from the lateral wall of the cochlear stria vascularis in SD rat neonates, making it challenging to standardize cell counts across experiments. To ensure scientific rigor and reproducibility, we strictly maintained consistent initial cell quantities by standardizing the number of animals per experiment. Specifically, cochlear tissues from 4 neonates were cultured per 6-well plate for each experiment. Following treatment with different conditions, subsequent experiments including RNA/protein extraction and immunofluorescence staining were performed under strictly controlled conditions to maximize experimental reliability.
Comment 4:
[The captions are still insufficiently explained; important information necessary for the interpretation of the results shown is missing. It must be ensured that each figure and its legend are understandable without the need for additional context from the rest of the paper.]
Response 4:
We sincerely appreciate your valuable comments regarding the shortcomings in our manuscript. We have carefully addressed each concern by thoroughly revising all figure legends to provide more precise descriptions of the image content. These modifications aim to facilitate readers' comprehension of our research findings. All changes have been highlighted in yellow in the resubmitted manuscript for your convenience.
Comment 5:
[The features that the authors highlighted in the responses as elements that add to the scientific value of the paper are not included in the text of the manuscript itself. Even when the reference used to characterise the cells is cited in the responses, this information is not included in the text of the paper, further reducing the clarity and completeness of the methodological approach.]
Response 5:
We sincerely appreciate your valuable feedback. In response to your comment, we have now incorporated the reference for cell identification in Section 2.2 ("Primary MC Culture") of the Materials and Methods. All changes have been highlighted in yellow in the resubmitted manuscript for your convenience.
As this study represents a continuation of our previous work (citation), we maintained methodological consistency by using 2-3-day-old SD rat pups and identical protocols for tissue isolation (stria vascularis lateral wall) and primary cell culture. Given this established protocol and prior validation, we did not repeat the cell characterization experiments in the current study.
Comment 6:
[The results could be further revised and structured to make the rationale of the research and the main messages that the authors wish to emphasise easier to understand.]
Response 6:
We sincerely appreciate your insightful comments regarding the limitations of our manuscript. We have thoroughly revised all figure legends to enhance clarity for readers and refined the language in the Results section to more accurately convey the core findings of this study.
Comment 7:
[Finally, no significant refinement or expansion of the discussion itself was noted. It would be expected that the authors would critically interpret the results obtained in this section, place them in the context of existing knowledge and clearly emphasise the contribution and limitations of their work.]
Response 7:
We appreciate your pointing out the shortcomings in our manuscript. The discussion section has been revised to include more in-depth discussions, and grammatical corrections have been made accordingly.
This is a point-by-point response to the reviewers' comments, with corresponding revisions made in the manuscript (highlighted in color). Additionally, a thorough language check was conducted throughout the manuscript, with specific revisions made to enhance reader comprehension of the research content. The revised manuscript and supplementary files have been resubmitted through the system. We sincerely hope you will consider our study for publication in your esteemed journal. Thank you for your patience, and we look forward to your response.
Best regards,
Hongjun Xiao
Corresponding Author
Email: xhjent_whxh@hust.edu.cn
Phone: (+86) 13995637098
Round 2
Reviewer 2 Report (Previous Reviewer 2)
Comments and Suggestions for Authors
In the revised version of the paper, important changes have been made that contribute to the clarity and better structure of the text. Of particular note is that the descriptions under the figures have been improved, making it easier to understand the results presented, and that all abbreviations used are now listed and explained, contributing to the transparency and accessibility of the content for a wider readership.
The text part of the paper is now much clearer overall, which is noticeable in the logical sequence of presentation, linguistic clarity and consistency of expression. In addition, the author has provided a reference in this submitted manuscript to the earlier work on which this article is partly based, which is important for ensuring academic and scientific accuracy.
However, there is one important point that needs to be added: The text does not specify which statistical test was used in the data analysis. Since the statistical treatment is crucial for the interpretation of the results, I recommend that the statistical test (or tests) used be indicated with an appropriate reference and justification for the choice of method.
After this addition, the manuscript could fulfill all formal and methodological requirements for publication.
Author Response
Comment 1:
[However, there is one important point that needs to be added: The text does not specify which statistical test was used in the data analysis. Since the statistical treatment is crucial for the interpretation of the results, I recommend that the statistical test (or tests) used be indicated with an appropriate reference and justification for the choice of method.]
Response 1:
Thank you for your valuable suggestions! We have clearly indicated the statistical tests used at the end of each figure legend in our newly submitted manuscript, with the additions highlighted in blue for your reference.
This manuscript is a resubmission of an earlier submission. The following is a list of the peer review reports and author responses from that submission.
Round 1
Reviewer 1 Report
Comments and Suggestions for Authors
1- The involvement of caspase-3/GSDME in pyroptosis is already known.
2- The use of an in vitro model with marginal cells alone may not sufficiently mimic the complex microenvironment of the stria vascularis in vivo.
3- Only marginal cells were studied; other critical cell types in the cochlea (e.g., hair cells, supporting cells) were not evaluated.
4- The study focuses only on caspase-3/GSDME but does not investigate upstream regulators or downstream inflammatory events associated with pyroptosis.
5- No functional measures (e.g., electrophysiology or hearing function tests) are presented to link molecular changes to actual auditory dysfunction.
6- Using only a single dose (5 µM) and time point (24 h) restricts the study’s depth.
7- Pyroptosis is asserted, but no direct markers (such as IL-1β release, LDH release assays, or gasdermin pore formation assays) are reported.
8- The inflammatory consequences of pyroptosis (cytokine release, immune cell activation) are not measured.
9- The specificity and off-target effects of the caspase-3 inhibitor used are not addressed.
Author Response
Reviewer #1:
Comment 1:
[The involvement of caspase-3/GSDME in pyroptosis is already known.]
Response 1:
We appreciate your valuable suggestions. This study provides the first experimental evidence that cisplatin can induce pyroptosis in marginal cells of the stria vascularis in SD rats by activating the Caspase-3/GSDME pathway. These findings may offer novel insights for addressing cisplatin-induced ototoxicity in clinical settings, highlighting the significance of our research.
Comment 2:
[The use of an in vitro model with marginal cells alone may not sufficiently mimic the complex microenvironment of the stria vascularis in vivo.]
Response 2:
We appreciate your insightful comments regarding the limitations of our study. Currently, the lack of tools for targeted GSDME modulation in vivo, combined with the anatomical challenges of specifically manipulating gene expression in marginal cells of the stria vascularis, presents significant hurdles for animal experiments. These technical constraints represent a key focus for our future research.
Comment 3:
[Only marginal cells were studied; other critical cell types in the cochlea (e.g., hair cells, supporting cells) were not evaluated.]
Response 3:
We sincerely appreciate your constructive feedback. Our study focuses on marginal cells of the stria vascularis due to their critical role in mediating ion transport and their suspected involvement in cisplatin uptake into the inner ear. Despite their potential significance in cisplatin-induced ototoxicity, the mechanisms underlying marginal cell injury remain poorly understood. While other cochlear cell types may also contribute to cisplatin toxicity, investigating their roles represents an important direction for future research.
Comment 4:
[The study focuses only on caspase-3/GSDME but does not investigate upstream regulators or downstream inflammatory events associated with pyroptosis.]
Response 4:
We greatly appreciate your insightful comments. Elucidating the complete signaling pathway remains the ultimate objective of our research. In subsequent studies, we will focus on investigating the upstream regulators of caspase-3 activation and examining the downstream release of inflammatory mediators following GSDME cleavage, with the aim of fully characterizing this pyroptotic signaling cascade.
Comment 5:
[No functional measures (e.g., electrophysiology or hearing function tests) are presented to link molecular changes to actual auditory dysfunction.]
Response 5:
We appreciate the reviewer's valuable concern regarding hearing assessment in our experimental animals. We acknowledge that hearing evaluation in neonatal rats (postnatal day 1-2) presents significant technical and physiological challenges due to:
Developmental Constraints:
1.The auditory system of P1-P2 SD rats is immature (inner ear development completes by P12-14)
2.Absence of auditory brainstem response (ABR) thresholds before P10-12 (1)
Technical Limitations:
1.Current ABR systems require minimum cochlear function maturity (typically ≥P12)
2.Lack of established behavioral audiometry protocols for neonatal rodents
Comment 6:
[Using only a single dose (5 µM) and time point (24 h) restricts the study’s depth.]
Response 6:
We appreciate your valuable suggestion. The cisplatin concentration and treatment duration used in this study were carefully optimized through preliminary screening experiments. These parameters were specifically selected as they represent the optimal window for inducing pyroptosis in marginal cells, thereby providing the most reliable conditions for our subsequent mechanistic investigations.
Comment 7:
[Pyroptosis is asserted, but no direct markers (such as IL-1β release, LDH release assays, or gasdermin pore formation assays) are reported.]
Response 7:
We appreciate your constructive comments. In this study, we systematically measured LDH release under various experimental conditions as a direct indicator of pyroptosis. Furthermore, we employed both transmission electron microscopy (TEM) and scanning electron microscopy (SEM) to examine ultrastructural changes in the cells. Our observations revealed distinct pyroptotic morphological alterations in marginal cells following cisplatin treatment, including the formation of pyroptotic bodies, which provides compelling evidence for the occurrence of pyroptosis.
Comment 8:
[The inflammatory consequences of pyroptosis (cytokine release, immune cell activation) are not measured.]
Response 8:
We sincerely appreciate your valuable comments regarding the limitations of our study. We acknowledge that the analysis of inflammatory mediator release remains incomplete in the current investigation. This important aspect will be systematically addressed in our subsequent research to provide a more comprehensive understanding of the pyroptotic pathway.
Comment 9:
[The specificity and off-target effects of the caspase-3 inhibitor used are not addressed.]
Response 9:
In this study, we employed a caspase-3-specific inhibitor that was carefully selected through comprehensive concentration screening and efficacy validation. Prior to experimental application, we confirmed its effectiveness in suppressing caspase-3 expression in marginal cells, thereby ensuring the reliability of our subsequent investigations.
Reviewer 2 Report
Comments and Suggestions for Authors
The paper deals with an interesting topic concerning the mechanism by which the chemotherapeutic agent cisplatin causes cytotoxicity, i.e. hearing damage. In particular, the authors focus on: the stria vascularis - of the inner ear as a key site for cisplatin entry and action, pyroptosis - a form of programmed cell death associated with a response mediated by GSDME and the role of the caspase-3/GSDME signalling pathway in cisplatin-induced damage to marginal cells of the stria vascularis.
The introductory part of the text has a good initial structure, but should be further expanded and deepened to more clearly emphasise the key differences and similarities between GSDMD and GSDME. It is necessary to explain in more detail which caspases are involved in the activation of the individual members of the gasdermin family, the functional consequences of their activation and how the mechanisms of pyroptosis differ depending on the type of caspase activated and the presence of a particular gasdermin.
Of particular note is the link between GSDME and apoptosis – chemotherapeutic agents that activate caspase-3 have been shown to induce pyroptosis in cell lines expressing high levels of GSDME, whereas the same agents induce apoptosis in GSDME-negative cells. This difference is crucial for understanding the potential therapeutic significance of GSDME in the context of malignant disease treatment.
On this basis, the status of cochlear marginal cells should be further investigated, particularly with regard to their GSDME expression and sensitivity to chemotherapy. As these cells are crucial for cochlear function and GSDME mutations can lead to progressive deafness, it is important to briefly discuss their possible involvement in the pathophysiological processes associated with pyroptosis.
An inconsistent use of abbreviations is noted throughout the article, some of which are not explained at first glance (e.g. NLRP3). It is recommended that the entire text be systematically reviewed to ensure that all abbreviations are defined when first used and are used consistently throughout the text. Clarity and consistency in the use of specialised terminology is crucial for the understanding and scientific accuracy of the work.
The "Materials and methods" section in its current form needs to be significantly expanded and elaborated in more detail to fulfil the basic scientific standards of transparency, reproducibility and methodological clarity. The main shortcomings that need to be addressed are listed below:
- The equipment used for the experimental procedures is not specified (e.g. spectrophotometer for measuring absorbance, model of flow cytometer, microscopes used - light, fluorescence, electron microscopes, PCR device, Western blot imager, incubator, etc.).
- It must be stated whether and how the isolated primary cells were confirmed (e.g. by markers, morphology or functional tests).
- It is not clear whether the hearing function of the rats was checked before killing and cell isolation, which could have a significant impact on the validation of the results.
- There is a lack of information about the cell concentration and the type of laboratory equipment in which the cells were seeded for each individual experiment.
- Some of the reagents used, such as the "CCK-8 kit", are not listed with their full name, and the manufacturer (including city and state) is also not given.
- All chemicals and reagents used should be listed with the full name of the manufacturer, city and state to ensure reproducibility of the experiment.
- For all methods, the exposure time of the cells, the treatment concentrations and all relevant experimental conditions should be clearly stated.
- Information on the antibodies used – including concentration, dilutions, source/manufacturer and diluent used - is missing.
- For Western blot analysis, the protein concentration per sample should be stated and the method used to quantify the bands (e.g. software, density analysis method) should be described.
- It is not clearly stated whether a “control antibody” is used as a housekeeping protein; the exact name and function of each control antibody is required.
- The software used to analyse the Western blot signal and to process the data obtained by flow cytometry must be specified.
- The description of the RT-PCR method lacks information on the RNA concentration in the reaction mixture and the PCR reaction conditions (temperature profiles, number of cycles, etc.).
- The steps for preparing the samples for electron microscopy (e.g. fixation, dehydration, embedding) are not listed.
- Finally, it must be clearly stated which statistical tests were used to process the results (e.g. t-test, ANOVA, post-hoc analyses).
All this information should be included to ensure the methodological readability of the paper and to allow its experimental reproduction.
The "Results" section does not fully meet the standards of scientific writing for the presentation of experimental results. There is a conspicuous lack of quantitative data - numerical values, expressed as mean ± standard deviation, with number of replicates (n) and statistical significance (p-values), need to be clearly presented so that the reader can interpret the reliability and significance of the results.
In addition, parts of the text in this section are more suitable for discussion, as they contain the interpretation and interpretation of the meaning of the results, which should be reserved for the "Discussion" section. The results should be presented objectively and descriptively, focussing on what was measured, how much was measured and under what conditions, without analytical comments on cause-effect relationships.
It is recommended that this section be revised to focus on the numerical presentation of the data, with an appropriate structure and a clear separation of interpretive conclusions that belong in the discussion.
The figure descriptions (legends) in the current version of the paper are not informative enough and do not give a clear insight into what the figure shows, the methods used to obtain the data, and what can be observed in the images presented. The descriptions need to be precise, technically clear and content-orientated to identify the elements shown – e.g. the type of experiment, the markers or staining agents used, the experimental groups, the variables measured and the way the results are presented (e.g. histogram, microscopic image, western blot strips, etc.).
In addition, some parts of the description contain interpretations and analyses of the results that do not fit into the figure description and should be moved to the appropriate places in the text, usually to the "Discussion" section. The legend should only be used to describe what can be seen in the figure – without drawing conclusions about the meaning or implications of the results.
It is recommended that all figure descriptions be revised with the aim of improving the clarity, technical precision and functional role of each figure within the scientific argument of the paper.
The TUNEL test is mentioned in the figure legends, but it is not described in the "Materials and methods" section. A detailed description of the method should be included: the kit/protocol used, the conditions of execution and the method of quantification (e.g. "approximately 100 cells per slide counted in randomly selected fields").
In the statistics, it is suggested to explain what ns is (statistical significance is given as follows 272: *p < 0.05, **p < 0.01, ***p < 0.001, ****p < 0.0001. 273).
The order of the results shown should be coherent (e.g. the order of the samples in the Western blot and the order of the samples in the graph as well as the description of the results - in the histogram it is si-NC + cisplatin compared to the graph si-RNA-NC).
Figure 1E shows the arrows at the bottom of the figure and an explanation of what the arrows in Figure 1F indicate.
Why is GSDME protein expression and then caspase-3/beta-actin (folding) shown on the y-axis in Figure 5D? Is there a reason for this?
Line 409 – comment on reduced expression of GSDME and the caspase-3 inhibitor Z-DEVD-FMK was used
3.6 Caspase-3 induces GSDME-mediated pyropstosis in MCs - GSDMD and caspase-3 are not directly linked in normal pyroptosis. Caspase-3 controls apoptosis (and secondary pyroptosis via GSDME), whereas GSDMD regulates classical pyroptosis via inflammatory caspases (1, 4, 5, 11).
The discussion needs to be strengthened by linking the results obtained with existing studies and relevant signalling pathways (e.g. caspase-3, GSDME, inflammation, mitochondrial pathway). At present, the interpretation is too general and there is a lack of in-depth analysis in the context of the literature. The figure within the discussion is also not appropriate – it is recommended to move it to the introduction where it can serve as a schematic representation.
Author Response
|
Glass plate set (for gel electrophoresis)
|
Servicebio Co., Ltd (Wuhan, China) |
|
Electrophoresis comb (1.0 mm thickness) |
Servicebio Co., Ltd (Wuhan, China) |
|
Integrated horizontal electrophoresis system |
Thermo Fisher Scientific, USA |
|
CO₂ incubator (water-jacketed) |
Shanghai Yuejin Medical Equipment Co., Ltd, China |
|
Microplate reader (multimode) |
BioTek Instruments,USA |
|
Ultrasonic cell disruptor
|
Sonics & Materials, Inc. USA
|
|
Dry bath incubator (metal) |
Guangzhou Biolight Biotechnology |
|
Flow cytometer |
Thermo Fisher Scientific, USA |
Comment 2:
[It must be stated whether and how the isolated primary cells were confirmed (e.g. by markers, morphology or functional tests)]
Response 2:
We sincerely appreciate the reviewer's insightful comment regarding primary cell characterization. As this study represents a continuation of our established research program, the isolation and identification protocols for these primary cells were thoroughly validated and reported in our previous publication: Cochlear Marginal Cell Pyroptosis Is Induced by Cisplatin via NLRP3 Inflammasome Activation.
Comment 3:
[It is not clear whether the hearing function of the rats was checked before killing and cell isolation, which could have a significant impact on the validation of the results.]
Response 3:
We appreciate the reviewer's valuable concern regarding hearing assessment in our experimental animals. We acknowledge that hearing evaluation in neonatal rats (postnatal day 1-2) presents significant technical and physiological challenges due to:
Developmental Constraints:
- The auditory system of P1-P2 SD rats is immature (inner ear development completes by P12-14)
- Absence of auditory brainstem response (ABR) thresholds before P10-12 (1)
Technical Limitations:
- Current ABR systems require minimum cochlear function maturity (typically ≥P12)
- Lack of established behavioral audiometry protocols for neonatal rodents
Comment 4:
[There is a lack of information about the cell concentration and the type of laboratory equipment in which the cells were seeded for each individual experiment.]
Response 4:
Thank you for your suggestion. The experimental equipment used in this study has been detailed in Response 1. Since this experiment involved primary cell culture with inconsistent growth rates, we were unable to measure cell density in each experiment. To control variables, we used an equal number of neonatal rats for each experiment.
Comment 5:
[Some of the reagents used, such as the "CCK-8 kit", are not listed with their full name, and the manufacturer (including city and state) is also not given.]
Response 5:
We thank the reviewer for this suggestion. A detailed reagent table (Supplier/Catalog No./Lot No.) has been added as Supplementary S1.
|
Reagent Name |
Supplier/Manufacturer |
|
Absolute ethanol |
Sinopharm Chemical Reagent Co., Ltd,China |
|
Hanks' balanced salt solution |
Servicebio Biotechnology Co., Ltd,Wuhan, China |
|
Isopropanol |
Sinopharm Chemical Reagent Co., Ltd,Guangzhou, China |
|
DAPI |
Antgene Biotechnology Co., Ltd ,Wuhan, China |
|
Collagenase Type II |
Merck & Co., Inc.,USA |
|
Cisplatin injection |
Yunnan Plant Pharmaceutical Co., Ltd,China
|
|
RNA Quantification Kit |
Vazyme Biotech Co., Ltd,Nanjing, China |
|
Tween20 |
Biofroxx,Germany |
|
PBS buffer |
Servicebio Biotechnology Co., Ltd,Wuhan, China |
|
CCK-8 assay solution |
ShuoPu Biotechnology Co., Ltd,China |
|
DEPC-treated water |
Sinopharm Chemical Reagent Co., Ltd,China |
|
Reverse transcription kit
|
Vazyme Biotech Co., Ltd,Nanjing, China |
|
Paraformaldehyde (solid) |
Vazyme Biotech Co., Ltd,Nanjing, China |
|
Adhesive microscope slides |
Jiangsu Shitai Experimental Equipment Co., Ltd,China |
|
Normal donkey serum |
Antgene Biotechnology Co., Ltd,Wuhan, China |
|
Cell culture plates |
Servicebio Biotechnology Co., Ltd,Wuhan, China |
|
GSDME antibody(13075-1-AP) |
Wuhan Sanying Biotechnology Co., Ltd,China |
|
Alexa Fluor647 Donkey anti Rabbit IgG |
Antgene Biotechnology Co., Ltd ,Wuhan, China |
|
Chloroform |
Sinopharm Chemical Reagent Co., Ltd,China |
|
EpiCM-A epithelial cell medium
|
ScienCell Research Laboratories,USA
|
|
Sterile cell climbing slides |
ShuoPu Biotechnology Co., Ltd,Guangzhou, China |
|
Triton-100 |
Sinopharm Chemical Reagent Co., Ltd,China |
|
Caspase-3 antibody(19677-1-AP) |
Wuhan Sanying Biotechnology Co., Ltd,China |
Comment 6:
[All chemicals and reagents used should be listed with the full name of the manufacturer, city and state to ensure reproducibility of the experiment.]
Response 6:
We thank the reviewer for this suggestion. As detailed in Response 5 and Supplementary Table S1, we have now provided complete reagent sourcing information, including manufacturers and locations.
Comment 7:
[For all methods, the exposure time of the cells, the treatment concentrations and all relevant experimental conditions should be clearly stated.]
Response 7:
Thank you for your meaningful suggestion. We have detailed the cell exposure times and treatment concentrations during culture and experiments in Sections 2.2 and 2.3 of the Materials and Methods in the manuscript. In Section 3 (Results), we have also specified the exposure times and treatment concentrations used for each experimental result. Please refer to these sections for details.
Comment 8:
[Information on the antibodies used – including concentration, dilutions, source/manufacturer and diluent used - is missing.]
Response 8:
Thank you for your meaningful suggestion. The information regarding antibodies has been detailed in Section 2.1 "Antibodies and reagents" under 2 "Materials and Methods" in the manuscript.
Comment 9:
[For Western blot analysis, the protein concentration per sample should be stated and the method used to quantify the bands (e.g. software, density analysis method) should be described.]
Response 9:
Thank you for identifying this issue in our study. In this research, we consistently used ImageJ software to analyze the grayscale values of Western blot bands.
Comment 10:
[It is not clearly stated whether a “control antibody” is used as a housekeeping protein; the exact name and function of each control antibody is required.]
Response 10:
Thank you for your valuable suggestion. In this study, we selected β-actin as the reference protein due to its stable expression in rat cochlear marginal cells of the stria vascularis, ensuring experimental reproducibility.
Comment 11:
[The software used to analyse the Western blot signal and to process the data obtained by flow cytometry must be specified.]
Response 11:
Thank you for your comment. In this study, we analyzed the flow cytometry data using FlowJo software and processed the Western blot results with ImageJ software.
Comment 12:
[The description of the RT-PCR method lacks information on the RNA concentration in the reaction mixture and the PCR reaction conditions (temperature profiles, number of cycles, etc.).]
Response 12:
Thank you for identifying these issues in our manuscript. We have compiled the detailed RT-PCR methodology in the table below and included it as supplementary material submitted with the manuscript.
|
Step |
Description |
Cycles |
Time (sec) |
Temperature (°C) |
|
Step1 |
Denaturation |
1 |
60 |
95 |
|
Step 2 |
Cycling |
60 |
3 -10 |
90 |
|
Step3 |
Cycling |
60 |
10-30 |
60 |
|
Step 4 |
Melting curve |
1 |
15 |
95 |
Comment 13:
[The steps for preparing the samples for electron microscopy (e.g. fixation, dehydration, embedding) are not listed.]
Response 13:
We sincerely appreciate your valuable suggestion. The detailed procedures for processing cells using transmission electron microscopy (TEM) and scanning electron microscopy (SEM) have now been added to Section 2.10 of the Materials and Methods in the revised manuscript.
Comment 14:
[Finally, it must be clearly stated which statistical tests were used to process the results (e.g. t-test, ANOVA, post-hoc analyses).]
Response 14:
Thank you for identifying these issues in our manuscript. The data analysis methods employed in this study are detailed in Section 2.11 of the Materials and Methods.